

# Effects of carbodiimide combined with ethanol–wet bonding pretreatment on dentin bonding properties: an *in vitro* study

Xiaoxiao You, Long Chen, Jie Xu, Sihui Li, Zhenghao Zhang and Ling Guo

Department of Prosthodontics, The Affiliated Stomatological Hospital of Southwest Medical University, Luzhou, China

## ABSTRACT

**Purpose**. This study evaluated the combined effects of Carbodiimide (EDC) and ethanol–wet bonding (EWB) pretreatment on the bond strength and resin–dentin surface.

**Methods**. Phosphoric acid-etched dentin specimens were randomly divided into five groups based on the following pretreatments: deionized water (control), EWB, 0.3M EDC in water (EDCw), EDC water solution combined EWB (EDCw + EWB), and 0.3M EDC in ethanol (EDCe). A scanning electron microscope (SEM) was used to observe the morphology of collagen fibrils on the demineralized dentin matrix in each group after pretreatment. The adhesives Prime & Bond NT (PB) (Dentsply De trey, Konstanz, Germany) or Single bond 2 (SB) (3M ESPE, St. Paul, MN, USA) was applied after pretreatments, and a confocal laser scanning microscope (CLSM) was used to evaluate the quality of resin tags. The degree of conversion (DC) of the adhesive was investigated by Fourier transform infrared spectroscopy (ATR-FTIR). The dentin was first bonded with resin and bathed in water at 37 °C for 24 h. Half of them were subjected to 10, 000 cycles in a thermocycler between 5 °C and 55 °C before a microshear bond strength ($\mu$SBS) test. The statistical methods were Analysis of Variance (ANOVA) and a Tukey *post hoc* test at $\alpha = 0.05$.

**Results**. The $\mu$SBS was significantly affected by pretreatments ($p < 0.001$), adhesives ($p < 0.001$), and aging conditions ($p < 0.001$) as revealed by the three-way ANOVA. The EDCw, EDCw + EWB, and EDCe groups significantly increased the $\mu$SBS; the EDCw + EWB and EDCe groups produced the highest $\mu$SBS. In the EDC-containing groups, the SEM showed at the collagen fibrils in the dentin matrix formed a three-dimensional network structure in the tubules after cross-linking into sheets, and the hybrid layer formed thicker resin tags under a CLSM. In the EDC-containing groups, the CLSM observed an increase in the length of resin tags. PB showed a higher DC and bonding strength than SB, and the five pretreatment groups tested did not affect the DC of the two adhesives.

**Conclusions**. In etch–and–rinse bonding system, EDC combined with EWB pretreatment can improve the quality of the hybrid layer and enhance the mechanical properties of demineralized dentin matrix. Pretreatment with EDC–ethanol solution may be a new clinically friendly option for enhancing dentin bonding durability.

Corresponding author
Ling Guo, 372083745@qq.com

## INTRODUCTION

The rapid development of the dentin bonding technique has simplified clinical procedures and greatly shortened clinical treatment times (*Van Meerbeek et al., 2020*). However, clinical failures, such as marginal microleakage, secondary caries, and restoration shedding suggest that resin–dentin bonding has poor durability (*Breschi et al., 2018*; *Yi et al., 2019*).

The quality of the hybrid layer is the key to the durability of resin-dentin bonding (*Tran & Tran, 2021*). The hybrid layer is a new hard tissue formed by the infiltration of adhesive monomers into the demineralized dentin matrix. Ideally, the adhesive penetrates the matrix to establish chemical bonds, and wraps the demineralized collagen fibril meshwork within the matrix to form a micromechanical locking structure (*Trevelin et al., 2019*). Unfortunately, in practice, the hybrid layer is the weakest link in the resin–dentin interface (*Maassen, Wille & Kern, 2021*; *Mazzoni et al., 2017*; *Spencer, Wang & Katz, 2004*). The reason for this weakness is that the demineralized dentin needs to be kept sufficiently moist to prevent the collagen fibril from collapsing and promote the penetration of the adhesive monomer (*Hammal et al., 2021*; *Kanca 3rd, 1992*; *Tsujimoto et al., 2019*). However, it is difficult to control the water humidity in clinical operation, and excessive water will interfere with the complete polymerization of the resin (*Yi et al., 2019*). Moreover, this phenomenon is more serious in etch-and-rinse systems because the etch-and-rinse technique uses an acid etchant to remove the smear layer and form a 5–10 um layer of demineralized dentin. The demineralized collagen fibril network is suspended in 70% volume of water after rinsing (*Jee et al., 2016*; *Tran & Tran, 2021*). The adhesive monomer has suboptimal infiltration and cannot fully infiltrate into wet demineralized dentin, leaving an incomplete infiltrated area containing denuded collagen fibrils at the bottom of the hybrid layer (*Breschi et al., 2018*; *Jee et al., 2016*). Since there is no separate etching step in the self-etching system, the adhesive co-monomer can demineralize and infiltrate into the dentin matrix at the same time, which reduces the difference between the demineralization and the resin infiltration depths, and it can form a hybrid layer with more homogenous resin infiltration (*Breschi et al., 2018*). The main reasons for the failure of resin–dentin bonding are the hydrolysis of bonding resin and the enzymatic hydrolysis of collagen fibrils by endogenous matrix metalloproteinases (MMPs) (*Breschi et al., 2018*). Several targeted strategies have been proposed to improve the durability of dentin bonding to protect the quality of the hybrid layer, such as using chlorhexidine to inhibit the activity of enzymes (*Shen et al., 2020*), remineralization of hybrid layers (*Braga & Fronza, 2020*), calcium-chelation dry bonding (*Mai et al., 2017*), cross-linking agents, and removal of the unbound/residual water within the hybrid layer (*Breschi et al., 2018*). However, only a few protocols have actually been considered for clinical use.

The ethanol wet bonding (EWB) technique is one of the most effective strategies for improving the formation of the hybrid layer (*Stape et al., 2021*). On the one hand, ethanol can replace the water in interfibrillar and intrafibrillar spaces while maintaining the three-dimensional shape of collagen fibrils, promoting a greater number of demineralized collagen fibrils and the adhesive wrapping of active sites of endogenous proteases (*Pashley et al., 2011*). On the other hand, the adhesive monomers are more soluble and permeable

in ethanol than in water, so more adhesive monomers are guided into the hybrid layer to reduce water absorption over time (*Yesilyurt et al., 2015*). However, the standard EWB technique is complex and takes 3–4 min, limiting its clinical use (*Ayar, 2016*); detrimental effects on the durability of composites *in vivo* have been reported (*Kuhn et al., 2015*). Simplifying the technique may lead to clinical applications and aid in dentin bonding (*Pashley et al., 2011*; *Van Meerbeek et al., 2020*).

Demineralized collagen fibril is an important scaffold for adhesive monomers and the underlying mineralized dentin to form a hybrid layer (*Yang et al., 2020*). Its three-dimensional structure is prone to collapse due to the interference of various physical and chemical factors in the bonding process. (*Breschi et al., 2018*; *Ryou et al., 2015*). Recent studies have shown that Carbodiimide (EDC), as a zero-length chemical cross-linker, directly conjugates carboxylates to primary amines in the collagen polypeptide chain without becoming part of the final crosslink (*Comba et al., 2019*; *Maravic et al., 2021*), and it enhances the mechanical properties of collagen through intermolecular and intramolecular crosslinking (*Hebling et al., 2015*; *Lopes et al., 2020*). In addition, EDC can cross-link matrix Metalloproteinases (MMPs) and inactivate the catalytic sites on MMPs by altering the three-dimensional conformation of MMP molecules, which inhibits MMPs and prevents the enzymatic hydrolysis of collagen fibrils (*Comba et al., 2019*).

Several studies that pretreated demineralized dentin with 0.3M EDC in water for 1 min have demonstrated the improved bonding strength of the resin–dentin interface (*Comba et al., 2019*; *Hardan et al., 2022*; *Maravic et al., 2021*; *Mazzoni et al., 2013*; *Mazzoni et al., 2014*). However, studies have shown that nearly half of the demineralized dentin the adhesive should infiltrate is still infiltrated by residual water during the process of dentin bonding (*Tjaderhane et al., 2013*). Residual water in the dentin matrix threatens the hybrid layer with hydrolysis and enzymatic hydrolysis, and resin–dentin bond strengths often decrease by 50%–60% in just 6–12 months (*Breschi et al., 2008*). Therefore, it is speculated that using EWB after EDC pretreatment to remove residual water in the dentin matrix may improve the formation quality of the hybrid layer.

Single bond 2 (SB) (3M ESPE, St. Paul, MN, USA) and Prime & Bond NT (PB) (Dentsply De trey, Konstanz, Germany) are fifth-generation two-step etch and rinse adhesives widely used in clinical practice. However, their high hydrophilicity and water sorption are major risks to the stability of bonding (*Sofan et al., 2017*). Moreover, this property of adhesives is considered to be the main degradation mechanism of resin–dentin when nanoleakage related to water absorption or hydrolysis occurs (*Van Meerbeek et al., 2020*). Measures should be taken to improve their adhesion stability.

Since EDC can be dissolved in both water and ethanol, the purpose of this study is to evaluate the effects of different pretreatment methods (EDC water solution (EDCw), EWB, EDC water solution combined EWB (EDCw + EWB), and EDC ethanol solution (EDCe)) on the immediate and aging bond strengths of two etch-and-rinse adhesives (SB or PB), and to make further exploration of collagen micromorphology, resin morphology of hybrid layer, and the degree of conversion.

The null hypotheses are that there are no differences between EWB and EDC pretreatments in (i) the collagen micromorphology and resin morphology of the hybrid

layer, (ii) the immediate and aging bond strengths of SB or PB treated resin–dentin interface, and (iii) the degree of conversion of SB and PB.

## MATERIAL AND METHODS

### Specimen preparation and pretreatment

This project was approved by the Human Ethics Committee of the Affiliated Stomatological Hospital of Southwest Medical University (20211129001), and the informed consent of patients was obtained in writing. After the periodontal film and dental calculus were removed, 130 non-carious extracted human molars were stored in 1% chloramine solution to prevent bacteria growth at 4 °C for no more than 6 months.

Roots of teeth were embedded in the mold using polymethyl methacrylate (LELE, Shanghai, China). According to the pretreatment methods, all the prepared teeth were randomly separated into five groups ($n = 26$), and the pretreatment method of each group is shown in Table 1.

### Scanning electron microscope analysis

Ten teeth were divided into five pretreatment groups. The occlusal enamel was removed with a low-speed diamond saw blade to expose the superficial dentin, which was then cut into 1-mm-thick dentin slabs perpendicular to the long tooth axis. A 600-grit silicon carbide (SiC) paper polished the occlusal surface of the dentin slabs for 60 s under running water to create a uniform smear layer. Subsequently, the dentin surfaces were etched with 35% phosphoric acid (Gluma, Hanau, Germany) for 15 s, rinsed for 20 s with water- spray, and blot-dried with paper. (Fig. 1a3). After pretreatment (Table 1, Fig. 1a4), specimens were immediately stored in a 2.5% buffered glutaraldehyde-based fixative solution (Yuanye, Shanghai, China) for 12 h at 4 °C. After dehydration in ethanol with increasing concentration gradients and spray-gold, the specimens were observed using an Inspect F50 (FEI Co., Thermo Fisher, USA) at magnifications of 10000× and 50000×.

### Confocal laser scanning microscopy

Twenty teeth were divided into ten groups according to pretreatment methods (Control, EWB, EDCw, EDCw + EWB, EDCe) and adhesives (SB or PB). After the occlusal enamel was removed, the specimens were cut into 1-mm-thick dentin slabs. The occlusal surfaces of the dentin slabs were acid-etched, rinsed thoroughly, and blot-dried. The specimens were pretreated in groups according to the necessary requirements (Table 1). 0.1% rhodamine B was added to the adhesive and fully mixed, then the adhesive was applied to the surface of dentin slabs and polymerized for 20 s (Fig. 1b5). The compositions in the adhesives are shown in Table 2. After dentin slabs were glued on glass slides, the bonding interface was observed under a confocal laser scanning microscope (CLSM) (TCS SP8, Leica Microsystems, Germany). CLSM Confocal laser scanning microscope images were obtained by z-steps of 1 mm, and optical sections were carried out from the bonding surface to the disappearance of the rhodamine staining signal. The three-dimensional morphology of the bonding interface was reconstructed using software programs (LAS X 3.7.4, Leica Microsystems, Germany).

**Table 1  The operation method of five pretreatment groups.**

| Group | Treatment |
| --- | --- |
| Control | Apply deionized water to dentin surface for 1 min. |
| EWB | Apply 100% ethanol to dentin surface for 1 min. |
| EDCw | Apply 0.3 M EDC water-solution to dentin surface for 1 min |
| EDCw+EWB | Apply 0.3 M EDC water-solution to dentin surface for 1 min, then apply 100% ethanol for 1 min. |
| EDCe | Apply 0.3 M EDC ethanol-solution to dentin surface for 1 min. |

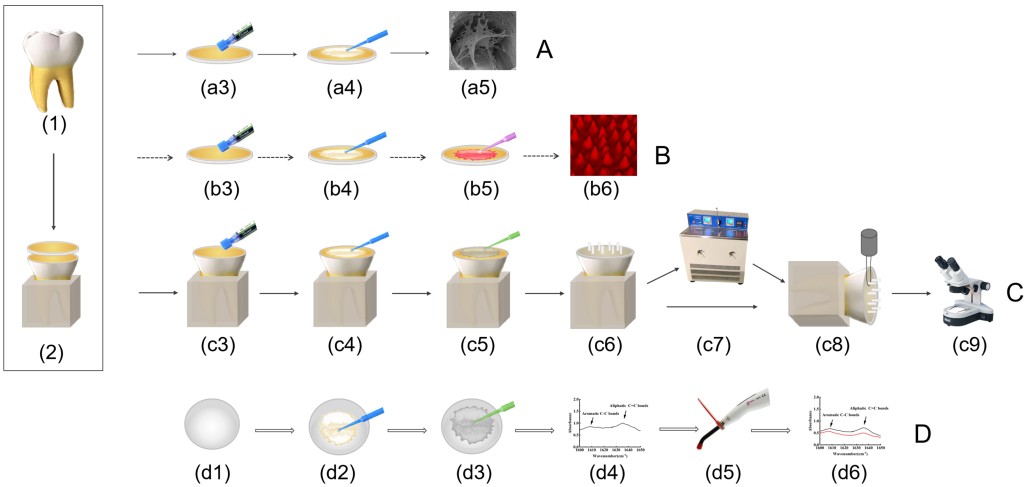

**Figure 1  Schematic drawing showing specimen preparation and testing.** (A) SEM analysis; (B) CLSM analysis; (C) μSBS test and failure mode analysis; (D) degree of conversion analysis. (1) The periodontal film and dental calculus of extracted molars were removed. (2) Roots of teeth were embedded, occlusal enamel was removed, specimens were cut into one mm thick dentin slabs. (a3) (b3) (c3) the dentin surfaces were etched with 35% phosphoric for 15 s, rinsed for 20 s and blot-dried with paper. (a4) (b4) (c4) (d2) Pretreatment was carried out according to the groups. (a5) Collagen fibrils morphology was observed by SEM. (b5) Apply adhesive which mixed with rhodamine B. (b6) the quality of resin tags were evaluated by CLSM. (c5) (d3) Apply adhesive. (c6) Eight tygon tubes was filled with composite resin. (c7) Half of the specimens were cycled 10,000 times in a thermocycler. (c8) Immediate and aging specimens were subjected to microshearing. (c9) The surfaces of fractured specimens were examined under a stereo microscope. (d4) FTIR analysis. (d5) Light-polymerized. (d6) FTIR analysis after light- polymerized.

## Microshear bond strength (μSBS) test

One hundred teeth were divided into five groups ($n = 20$ in each group) and pretreated as previously described. The specimens in each group were divided into two subgroups according to the adhesives used ($n = 10$ in each subgroup). After they were gently air dried, the pretreated dentin surface of each specimen (Figs. 1c3–1c4) was applied with adhesive according to the manufacturer's request (Fig. 1c5), which was polymerized by using a LED light-curing lamp (Woodpecker, Guangxi, China) for 20 s. Eight transparent tygon tubes with an internal diameter of one mm and a height of two mm were placed on each dentin's surface ensuring that the contact surface was fitted tightly (*Zhang et al., 2022*). Each tubule

**Table 2  Composition of adhesives used in this experiment.**

| Adhesives | Composition |
| --- | --- |
| Adper Single Bond 2<br>3M ESPE, St. Paul, USA<br>(SB) | Bis-GMA, HEMA, dimethacrylates, polyalkenoid acid copolymer, initiator, 34% water, ethanol |
| Prime & Bond NT<br>Dentsply De trey, Konstanz, Germany<br>(PB) | PENTA, UDMA, T-resin (cross-linking agent), D-resin, (small hydrophilic molecule), butylated hydroxitoluene, 4-ethyl dimethyl aminobenzoate, cetilamine hydrofluoride, acetone, silica nanofiller |

provided an area of 0.785 mm² for resin and dentin bonding. The tygon tubes were filled with composite resin (Filtek Z350; 3M, Saint Paul, MN, USA) and lightly polymerized for 20 s (Fig. 1c6). The tubes were carefully removed with a blade to expose the composite cylinders after the excess resin around the tubes was removed. A single trained operator performed all the bonding procedures.

After being stored in water at 37 °C for 24 h, half of the specimens were cycled 10,000 times between 5 °C and 55 °C in a thermocycler (TC-501F; Jiangsu, China). The exposure to each bath was 30 s, and the transfer time between baths was 10 s (Fig. 1c7). Subsequently, microshear bond strength tests (μSBS tests) were performed on both the immediate-bonding and aged-bonding subgroups. The mode of the universal testing machine (WDW-20; YINUO, Jinan, China) was set as the shear strength test, the test shape was set as round, and the test area was 0.785 mm². The 0.2 mm diameter orthodontic wire was bent into a circular arc, and the two ends of the orthodontic wire were embedded in methyl methacrylate to make a cutting tool. The specimen and shear tool were respectively fixed in the jig of the universal testing machine so that the resin column was perpendicular to the direction of the shear force (Fig. 1c8). After confirming that the orthodontic wire was wrapped around the base of each composite resin cylinder, all specimens were loaded at one mm/min until they fractured. The μSBS values (MPa) were calculated by dividing the force at failure (N) by the interface area (mm²). All data was recorded, including specimens with premature debonding; however, those values were not statistically analyzed because premature failures accounted for fewer than 3% of the total tested specimens (*Peng et al., 2020*). The average value of all resin cylinders on the dentin surface was used as the bond strength of each tooth.

## Failure mode analysis

After the μSBS test, the surfaces of fractured specimens were examined under a stereo microscope (Motic, Carlsbad, CA, USA) at 40× magnification to determine the failure mode (Fig. 1c9). When the fracture occurred at the resin–dentin interface, it was recorded as an adhesive failure. Fractures occurring in dentin or resin were recorded as cohesive failures. When the fracture occurred at the resin–dentin interface and included the cohesive failure of some adjacent substrates, it was recorded as a mixed failure. Fracture mode data was shown with GraphPad Prism 8.0 software (GraphPad Software, La Jolla, CA, USA).

## Degree of conversion analysis

Fourier transform infrared spectroscopy (FTIR) (IRAffinity-1S, Shimadzu, Japan) was used to analyze the conversion of the pretreated adhesive (SB or PB) (*Li et al., 2020*). Pretreatments were performed before adhesives were applied to potassium bromide tablets (Figs. 1d1–1d3). The absorption spectra aromatic C-C characteristic peaks (1608–1610 cm$^{-1}$) and aliphatic C $=$ C absorption peaks (1635–1640 cm$^{-1}$) of the adhesives before and after polymerization were measured by FTIR according to the standard method (*Yoshida et al., 2005*) (Figs. 1d4–1d6). The conversion rate of the adhesive was obtained by the following equation (*Al-Hamdan et al., 2020*; *Alhenaki et al., 2021*; *Yoshida et al., 2005*):

$$DC = \left\{ 1 - \frac{polymerized\left[\frac{Abs(C=C)}{Abs(C-C)}\right]}{unpolymerized\left[\frac{Abs(C=C)}{Abs(C-C)}\right]} \right\}.$$

## Statistical analysis

After validating the normality (Shapiro–Wilk test) and equal variance assumptions (Levene's test) of the data, the μSBS data was analyzed using a three-way (variables: different pretreatments, adhesives and aging) analysis of variance (ANOVA) and the Tukey *post hoc* test. Because the data of the degree of conversion (DC) was normally distributed and homoscedastic, they were evaluated by a two-way ANOVA, and the Tukey *post hoc* test was used to analyze the statistical differences between the two subgroups. SPSS 23.0 (Armonk, NY, USA) was used to analyze all data ($\alpha = 0.05$).

# RESULTS

## Scanning electron microscope analysis

Representative SEM images of the pretreated dentin matrix are illustrated in Fig. 2. After phosphoric acid etching and rinsing, dentin tubules and collagen fibrils were exposed. In the control and EWB groups, the collagen fibrils of the dentin matrix were few and sparse. In the EDCw, EDCw + EWB, and EDCe groups, it was observed that the collagen fibrils in the dentin matrix formed a three-dimensional network structure in the tubules after cross-linking into sheets.

## Analysis of the hybrid layer (CLSM)

Representative CLSM images of the morphology of resin tags from the groups tested are presented in Fig. 3. Generally speaking, the resin tags in the PB group were thicker, and the resin tags in the SB group were longer and thinner. The resin tags in the control group were sparse and short. In the ethanol treated groups, the resin penetrated deeper into the dentin tubules, and the length of the resin tags increased significantly. After EDC pretreatment, the resin tags were thicker and denser. The EDCe group showed the longest and densest resin tags.

## Microshear bond strength

The data of μSBS obtained from each group are presented in Table 3. The results of three-way ANOVA showed that the interactions among the three variables (pretreatments,

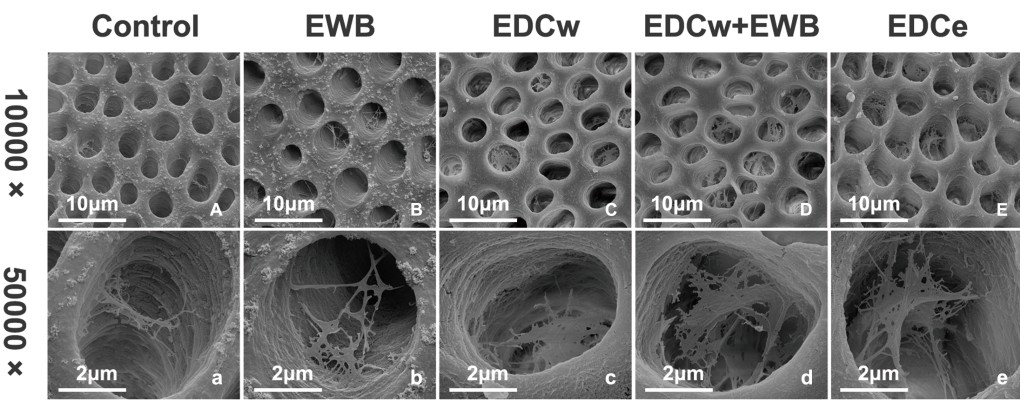

**Figure 2 Representative SEM images of dentin matrix by different pretreatments at a magnification of 10,000 × (A–E) and 50,000 × (a–e).** (A–B and a–b) Collagen fibrils of the dentin matrix were few and sparse. (C–E and c–e) Collagen fibrils formed a three-dimensional network structure in the tubules after cross-linking into sheets.

adhesives and aging) were not statistically significant. The adhesive and aging had an interactive effect ($F = 6.227$, $p = 0.015$). For all groups, the adhesive of PB significantly increased the bond strengths compared with SB ($F = 9.25$, $p < 0.05$). The bond strength values significantly decreased after 10,000 thermal cycles ($F = 49.328$, $p < 0.001$), different pretreatment significantly affected dentin bond strength ($F = 14.746$, $p < 0.001$). The pretreatments of EDCw ($p < 0.05$), EDCw + EWB ($p < 0.001$), and EDCe ($p < 0.001$) all significantly improved the bonding strength compared with the control group. Although the value of the bonding strength in the EWB group was higher than that in the control group ($p < 0.05$), there was no significant difference. In addition, the average bonding strength in the EDCe and EDCw + EWB groups was significantly higher than that in EDCw and EWB groups ($p < 0.05$).

## Failure mode analysis

The failure modes of each group are shown in Fig. 4. Adehesive failure accounted for the largest proportion of failures. After 10,000 thermal cycles, the adhesive failure of each group increased. The failure mode distribution was almost the same in the control, EWB, and EDCw groups, while the EDCw + EWB and EDCe groups showed a higher percentage of mixed and cohesive failures.

## Degree of conversion

The representative FTIR absorption spectra images of different adhesives and pretreatments are shown in Fig. 5. The results indicated that different pretreatments did not hamper the polymerization of SB and PB ($F = 0.524$, $p > 0.05$), while the DC of PB was significantly higher than SB ($F = 12.14$, $p < 0.05$). There was no interaction between adhesives and pretreatments ($F = 1.935$, $p > 0.05$).

Peer J

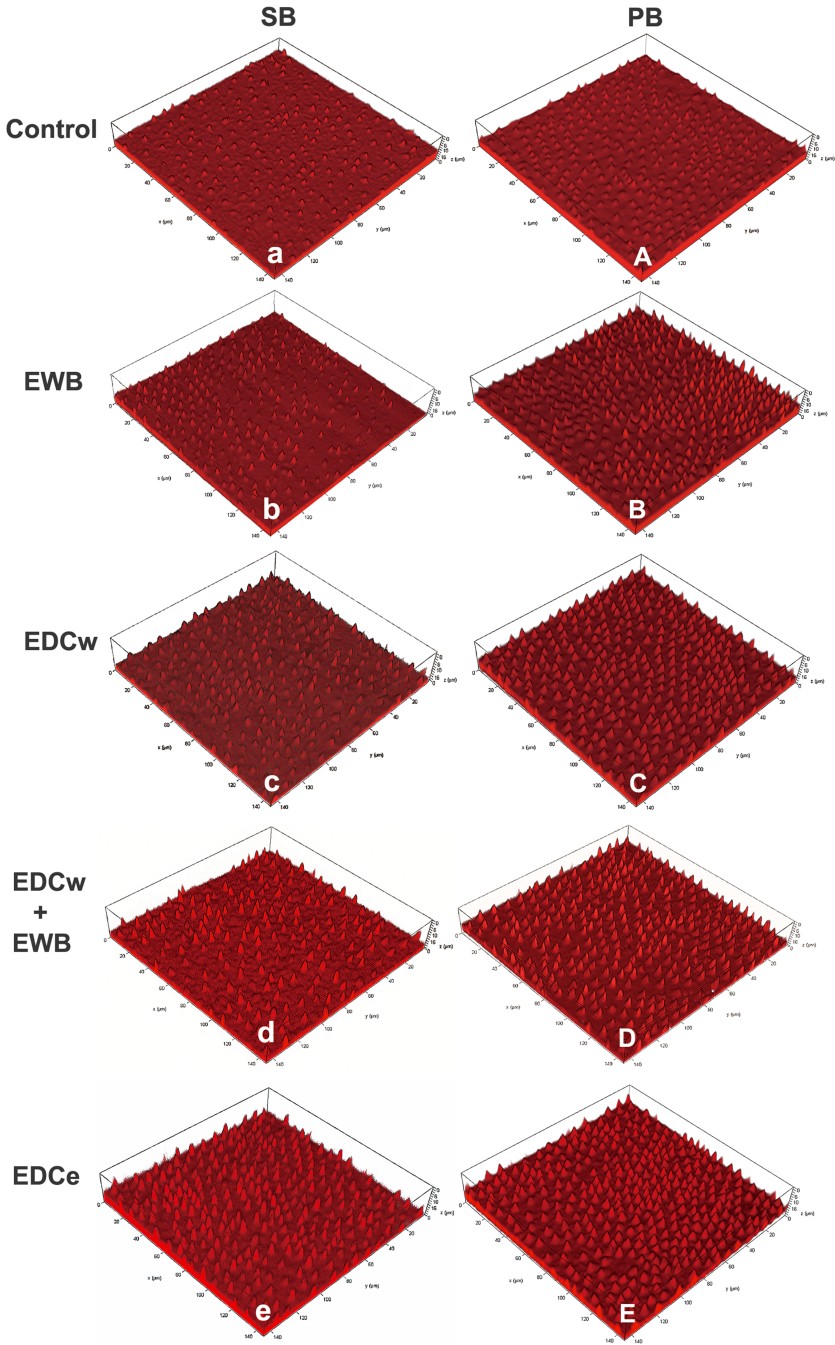

**Figure 3 Representative CLSM images of the resin–dentin interface created by the etch-and-rinse adhesive system SB (Single Bond 2) and PB (Prime & Bond NT) in different pretreatment groups.** Rhodamine B-labeled adhesive excited red fluorescence under microscope, depicting morphology of bonding interfaces. The resin tags in PB group are thicker, and the resin tags in SB group are longer and thinner. The resin tags in the control group (A, a) were sparse and short. In the ethanol treated group (B, D, b, d), the resin penetrated deeper into the dentin tubules and the length of the resin tags increased significantly. After EDC pretreatment (B, D, E, b, d, e), the resin tags are thicker and denser. The EDCe group (E, e) showed the longest and densest resin tags.

**Table 3** **Microshear bond strengths in MPa (means ± standard deviations) of the different groups.**

| | Group | Control | EWB | EDCw | EDCw+EWB | EDCe |
|---|---|---|---|---|---|---|
| SB | Immediate | 20.89 ± 1.84[ABb] | 22.86 ± 1.97[ABab] | 24.65 ± 1.30[Aab] | 24.01 ± 2.81[Bab] | 25.73 ± 1.94[Aa] |
| | Aging | 18.6 ± 2.64[Bb] | 20.39 ± 2.51[Bab] | 20.50 ± 2.75[Bab] | 24.04 ± 2.62[Ba] | 24.41 ± 2.92[Aa] |
| PB | Immediate | 23.14 ± 1.34[Ab] | 25.9 ± 1.84[Aab] | 26.2 ± 2.69[Aab] | 28.62 ± 1.49[Aa] | 26.97 ± 2.83[Aab] |
| | Aging | 20.08 ± 1.63[ABb] | 20.9 ± 2.25[Bab] | 20.52 ± 2.02[Bab] | 24.39 ± 2.53[ABa] | 23.53 ± 1.89[Aab] |

**Notes.**
Different uppercase letters indicate significant difference in each row, and lowercase letters indicate significant difference between columns from the Tukey' test ($P < 0.05$).

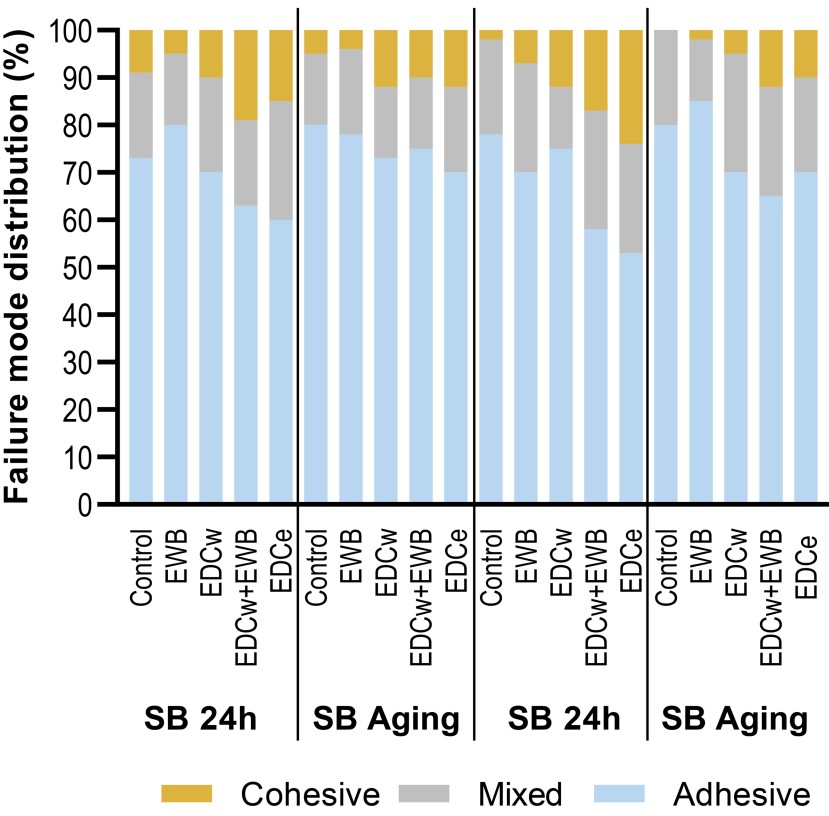

**Figure 4** **Failure modes distribution (%) for the groups tested under immediate or aging condition.**

# DISCUSSION

The formation quality of the hybrid layer is the main reason for the poor durability of dentin bonding (*Mazzoni et al., 2017*; *Tran & Tran, 2021*). The hybrid layer formed by the etch-and-rinse technique has exposed demineralized collagen fibrils at the bottom and is surrounded by rinse water, making it easier for the bonding interface to be hydrolyzed and enzymolized (*Mazzoni et al., 2017*). Although widely used in clinical practice, the fifth-generation adhesive of mixing hydrophilic and hydrophobic bonding monomers is not reliable in terms of bonding stability (*Sofan et al., 2017*; *Van Meerbeek et al., 2020*). To solve the problem of poor adhesive durability of the etch-and-rinse system, this study

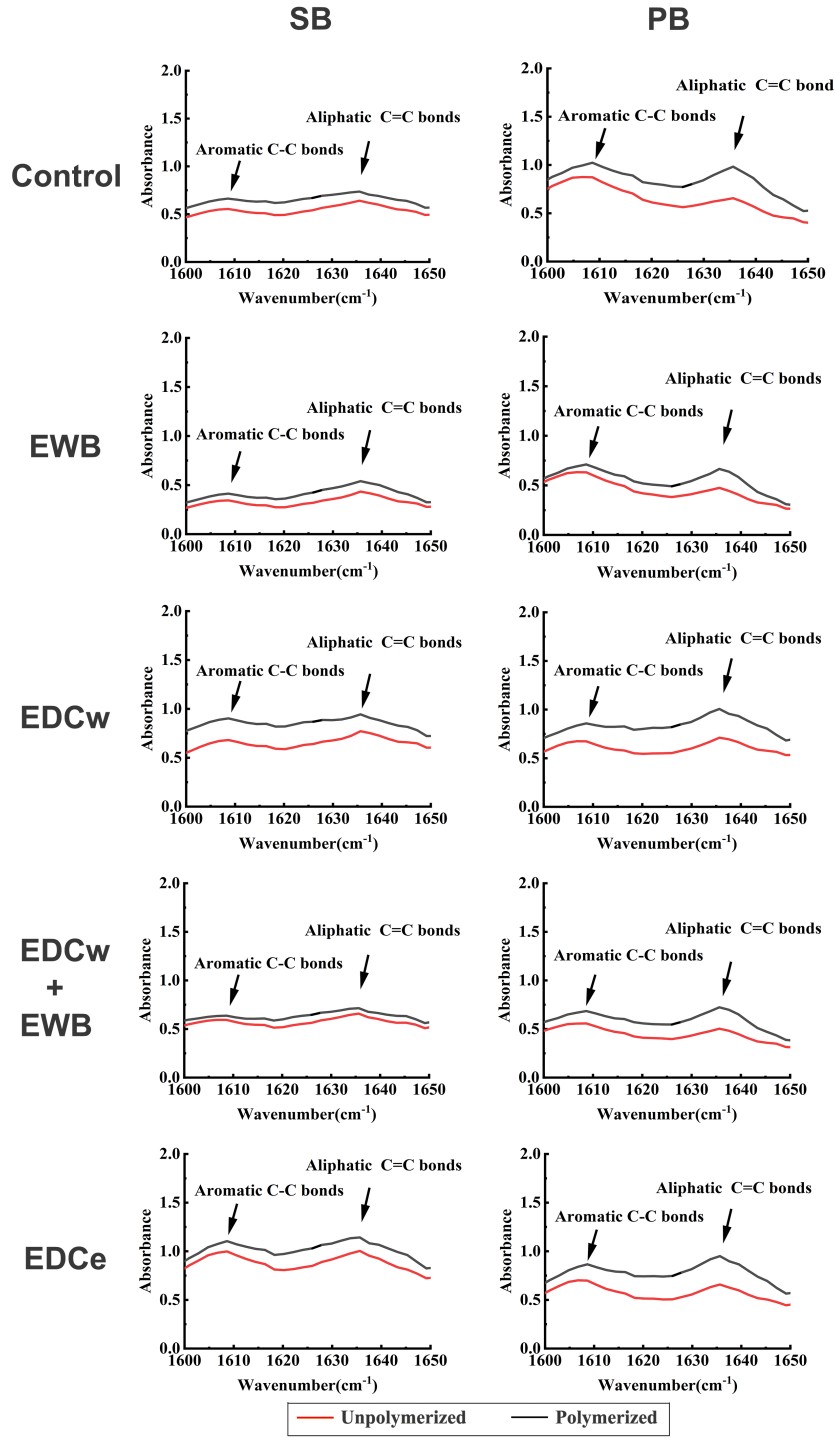

**Figure 5** **Fourier-transform infrared spectroscopy spectra of SB and PB for determining absorbance of $\gamma$ (C=C) and $\gamma$ (C-C) vibrations before and after light polymerization in each group.** The absorption bands of aromatic C-C are in the position of 1608–1610 cm$^{-1}$, the absorption bands of aliphatic C=C are in the position of 1635–1640 cm$^{-1}$. Unpolymerized, absorption spectra of unpolymerized adhesive; Polymerized, spectra of adhesive after light polymerization.

combined EWB, which can replace the excess water in the dentin matrix, with EDC, which promotes collagen cross-linking and inhibits MMPS activity. This paper evaluated whether this combination could show enhanced bonding properties.

The results showed that the collagen fibrils in the EDCw, EDCw + EWB, and EDCe groups formed a three-dimensional network structure after pretreatment, while the resin tags became longer after EWB pretreatment and denser after EDC treatment, so the null hypothesis of the first test was rejected. The μSBS results showed that different pretreatments, adhesives, and thermocycling had significant effects on the bonding strength of dentin, so the second hypothesis was also rejected. Since the DC of SB and PB are not affected by EWB and EDC pretreatments, the third null hypothesis was not rejected.

Carbodiimide treatment significantly improved the immediate bonding strength of resinous dentin, which was consistent with the results of previous studies (*Fernandes et al., 2021*; *Haught et al., 2016*). Carbodiimide enhanced the bonding strength by cross-linking with collagen, which enhanced the mechanical properties of collagen, and EDC has been reported to increase the stiffness of the demineralized dentin (*Bedran-Russo et al., 2008*; *Castellan et al., 2010*). The SEM verified the cross-linking of the collagen fibril. There were a greater number of collagen fibrils in the EDCw, EDCw + EWB and EDCe groups, which showed the tendency of fibrils connecting into a sheet, extending deeper into the dentin tubule, and finally forming a complex three-dimensional structure. In both SB and PB groups, the EDC treated groups showed thicker and denser resin tags in CLSM analysis than the control group after the adhesive was applied. This result suggests that the thicker collagen fibril scaffold can guide the adhesive to enter and fill the dentin tubules, creating closer contact with the underlying mineralized dentin and forming a higher quality hybrid layer, which increases the bonding strength.

In addition, although the bonding strength of all groups decreased after 10,000 thermocycling, the EDC-treated samples (EDCw, EDCw + EWB, and EDCe) maintained higher bonding strengths compared with the control group, which is consistent with the results of other studies (*Hardan et al., 2022*; *Maravic et al., 2021*; *Mazzoni et al., 2018*). The reason is speculated to be that covalent cross-links produced with EDC inactivate the active sites of dentin matrix metalloproteinases (*Maravic et al., 2021*; *Tjaderhane, 2015*), causing EDC-treated samples to undergo less hydrolysis and enzymatic hydrolysis, and maintain the integrity of the hybrid layer to a greater extent. Moreover, studies have found that even after aging, the use of EDC in dentin pretreatment can increase the elastic modulus of dentin and reduce the degradation rate of collagen (*Bedran-Russo et al., 2010*; *Maravic et al., 2021*).

The standard EWB dehydration protocol can enhance resin penetration and promote higher quality mixing layers (*Kuhn et al., 2015*), but the standard protocol is complex and takes 3–4 min, which is not suitable for clinical practice (*Ayar, 2016*). Therefore, the simplified EWB protocol is proposed. However, the ability of simplified EWB to improve bonding strength is still controversial. In several studies, 100% ethanol applied for 1 min replaced excess water in the dentin, while keeping collagen fibrils from collapsing and promoting resin monomer infiltration (*Ayar, 2016*; *S et al., 2010*; *Souza, DIN & Bresciani, 2018*). Unfortunately, in this study, although the bonding strength after EWB treatment

was observed to be 8.9% higher than that of the control group, no significant difference was shown. This result was consistent with other studies (*Nagpal, Manuja & Pandit, 2015*; *Yesilyurt et al., 2015*; *Yi et al., 2019*). The reason may be that, due to the high volatility of ethanol and the short 1 min treatment time of 100% ethanol, the ethanol did not penetrate deeper into the dentin matrix and did not completely replace the residual water (*Nagpal, Manuja & Pandit, 2015*; *Souza, DIN & Bresciani, 2018*). Another reason would be that because there is too much water in the dentin relative to the amount of ethanol applied, the ethanol concentration drops too low after 100% ethanol application to completely replace the water (*Jee et al., 2016*). However, in the CLSM study, longer resin tags were found in the EWB group than in the control group, and a similar phenomenon was observed in the EDCw + EWB and EDCe groups. This suggests that the ethanol displaces water from the dentin matrix, guiding the bonding components deeper into the dentin matrix to form a thicker hybrid layer (*Pashley et al., 2007*). In the dentin matrix, the more water removed, the more likely the polymer will enter the dentin matrix and form a good quality hybrid layer microlocking structure.

Compared with the EDCw group, the EDCe group showed better bonding strength. The CLSM observationfound that the thickness of resin protrusions in the EDCe and EDCw group was similar, but the resin tags in the EDCe group were longer and sharper than those in the EDCw group. This indicates that using water or ethanol as the solvent of EDC can produce similar crosslinking effects. However, ethanol as the solvent can also replace part of the water and promote the bonding monomers into the deeper dentin matrix.

The bonding strength of the EDCw + EWB and EDCe groups was the highest, and there was a significant difference compared with EDCw, but there was no difference between the two groups. On the one hand, this suggests that the bonding strength exhibited by the EDCw + EWB and EDCe groups resulted from the dual effect of crosslinking and EWB, which made them form denser and longer hybrid layers. The improvement of the quality of the hybrid layer was also confirmed by the increase in the proportion of mixing failure and cohesion failures in the EDCw + EWB and EDCe groups. On the other hand, it indicates that the order of crosslinking and EWB does not affect the bonding strength, and as shown in Figs. 2 and 3, the crosslinking degree of collagen fibrils and the resin morphology of the hybrid layer in the two groups were similar. Considering the convenience of its clinical operation, the pretreatment of EDC directly dissolved in ethanol may be a more suitable operation scheme to promote.

Out of the SB and PB adhesives, PB exhibited better bonding strength. This may be related to their monomer composition and concentration differences (*Yesilyurt et al., 2015*). Single Bond 2 contains the hydrophilic monomer HEMA, hydrophobic monomer bis-GMA and water, while PB contains the hydrophobic monomer UDMA, hydrophilic monomer PENTA and acetone. In addition, SB contains 34% solvent water, which leads to a reduction in the bonding strength of EWB (*Yesilyurt et al., 2015*). Prime&Bond NT uses acetone, which like ethanol, is an organic solvent, to displace residual water from the dentin matrix, and to transport the bonded monomers to the deep dentin matrix, where the acetone evaporates, and the functional monomers are left for subsequent polymerization (*Ekambaram, Yiu & Matinlinna, 2015*). Researchers have found that

acetone vapor pressure is higher than water and ethanoland can more effectively remove water from the demineralized dentin matrix (*Ekambaram, Yiu & Matinlinna, 2015*; *Van Landuyt et al., 2007*). In addition, acetone is less sticky and has better dentin permeability than ethanol, which can explain the higher bonding strength of PB than SB to a certain extent.

There is an important issue to be concerned about when the reagent is pretreated or added to the adhesive system before dentin bonding. When the reagent is added to the original adhesive solution, changes in the physical properties of the solution may significantly affect bonding efficacy and longevity (*Mendes et al., 2020*). To investigate the effect of ethanol and EDC pretreatments on the polymerization properties of adhesives, the DCs of the two adhesives were measured by FTIR (*Al-Hamdan et al., 2020*; *Alhenaki et al., 2021*; *Yoshida et al., 2005*). Since the peak intensity of aromatic C–C (1608–1610 cm$^{-1}$) does not change during the polymerization process, the absorbance changes of aliphatic C $=$ C absorption peaks (1635–1640 cm$^{-1}$) before and after the polymerization were calculated by using aromatic C–C as the internal standard to infer the DC from monomer to polymer (*Al-Hamdan et al., 2020*; *Moldovan et al., 2019*). There was no difference in DC values between the experimental and control groups, indicating that ethanol and 0.3 M EDC pretreatments did not affect the degree of polymerization of SB and PB, and the difference in the μSBS was not caused by the effects of EDC and ethanol on the adhesives.

There are potential clinical implications of this study. Carbodiimide cross-links with collagen fibrils to form a high-quality hybrid layer, tightly anchors the composite resin with mineralized dentin, and improves the immediate and aging bonding strength of dentin. Further protocols should be used to investigate the effects of EDCw and EDCe on dentin matrix proteases. More clinically applicable solutions should be further investigated, such as adding EDC directly to the adhesive and exploring an EWB technique that can improve the mechanical properties of dentin bonding and shorten the clinical operation time.

## CONCLUSIONS

Within the limitations of this study, the following conclusions were drawn:

1. EDC combined with EWB pretreatment effectively improved the immediate and aging bonding strength of demineralized dentin matrix, without affecting the degree of conversion of the etch-and-rinse adhesives.

2. EDC promoted the cross-linking of demineralized collagen fibrils; EWB promoted the penetration of adhesive resin to form longer resin tags. The combination of EDC and EWB more effectively improved the quality of the hybrid layer.

3. EDC–ethanol solution achieved the same enhancement of adhesive properties as EDC combined with EWB treatment. An EDC–ethanol solution may be a clinically friendly method to improve dentin bonding durability.

4. PB, which used only acetone as the solvent, had higher bonding strength and degree of conversion than SB which contains water in its composition.

## ACKNOWLEDGEMENTS

The authors thank Sichuan Institute of Atomic Energy for SEM supports.

### Funding

This work was supported by grant #S21008 of the Sichuan Medical Association (SMA) and grant #SNFZ20220004 of the Southwest Medical University Virtual Simulation project. The funders had no role in study design, data collection and analysis, decision to publish, or preparation of the manuscript.

### Grant Disclosures

The following grant information was disclosed by the authors:
Sichuan Medical Association (SMA): #S21008.
Southwest Medical University Virtual Simulation project: #SNFZ20220004.

### Competing Interests

The authors declare there are no competing interests.

### Author Contributions

- Xiaoxiao You conceived and designed the experiments, prepared figures and/or tables, and approved the final draft.
- Long Chen performed the experiments, authored or reviewed drafts of the article, and approved the final draft.
- Jie Xu performed the experiments, authored or reviewed drafts of the article, and approved the final draft.
- Sihui Li analyzed the data, prepared figures and/or tables, and approved the final draft.
- Zhenghao Zhang analyzed the data, prepared figures and/or tables, and approved the final draft.
- Ling Guo conceived and designed the experiments, authored or reviewed drafts of the article, and approved the final draft.

### Human Ethics

The following information was supplied relating to ethical approvals (i.e., approving body and any reference numbers):

This project was approved by the Human Ethics Committee (20211129001) of the Affiliated Stomatological Hospital of Southwest Medical University.

### Data Availability

The raw measurements are available in the Supplemental Files.

### Supplemental Information

Supplemental information for this article can be found online at http://dx.doi.org/10.7717/peerj.14238#supplemental-information.

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
