# Peer review of "Effects of carbodiimide combined with ethanol–wet bonding pretreatment on dentin bonding properties: an in vitro study"

_PeerJ, doi:10.7717/peerj.14238_

## Round 0.1 · original submission · Major Revisions

Dear Authors,

Your manuscript has been reviewed by external experts. The professional reviewers point out some major drawbacks that seriously diminish the research quality and scientific soundness. Before further consideration of your manuscript for publication in PeerJ, all the concerns raised by the referees should be addressed. Thank you very much.

Reviewer 1 ·

Basic reporting

Dear Authors,

This is a very important and interesting study. The purpose of the questions involved in the original research cover significant issues and attract many readers in the fields of dental materials and adhesive dentistry. However, the study has limitations in both the introduction and discussion sections where they need further improvements. Plus, the study involves lots of grammatical mistakes where they require correction by experts. The below points are suggestions to be considered for improving the quality of the research in the study and attracting valuable readers.

• The abstract contains an error which is EWB should be left as abbreviation (the full name was given on the purpose). Furthermore, the authors stated the adhesive without their brand, please correct. Please provide full name of ANOVA before abbreviation. Aging should be described in materials and methods. The authors stated SEM showed the degree of crosslinking was improved.... Please rephrase. The results section should be reformulated it's not well structured. Also, conclusion section needs reformulation.
• Introduction: L32-33: into demineralized dentin matrix+ Hybrid layer is formed not only by adhesive infiltration … please correct. L38: Full polymerization should be replaced by complete polymerization. L45: a number of strategies: please provide some of them + references. You should talk more about adhesives and the problems of etch-an-rinse adhesives then you talk about solution. Lack information about adhesives in this introduction. L62: Please add reference for this sentence. L62: EDC provide full name then abbreviation. L81: Put the name of adhesives
• Materials and methods: L97: Silicone carbide paper the abbreviation
• Lack of sample size calculation
• SEM analysis you should describe more how you prepare and treat the specimens
• Failure mode please describe more
• Discussion shouldn’t directly start with the first hypothesis…. Please rearrange start with a background then start with rejection or acceptance.
• Please add more limitations

Experimental design

Research question well defined, relevant & meaningful. It is stated how research fills an identified knowledge gap.

Validity of the findings

Conclusion section needs reformulation.

Reviewer 2 ·

Basic reporting

• Please use updated and recent papers in the literature review to give more sense to the reader.
• The writing of the manuscript could be improved by an expert English writer.methodology needs major improvements.

Experimental design

Methodology needs major improvements.

Validity of the findings

Conclusions and results were well defined.

Additional comments

Dear Authors,

It is an excellent paper that can increase understanding by assessing the bond strength of novel strategies in adhesive dentistry. However, there are a few suggestions to improve it as follows:

• The abstract could be more comprehensive by focusing on important parts. Please rewrite the abstract and try to emphasize the significant parts.
• Keywords should be determined by the appropriate MeSH Terms in NCBI.
• The writing of the manuscript could be improved by an expert English writer.
• More information regarding the mechanical test must be provided, since the testing mode of the universal testing machine, was not specified.
• Please indicate sizing in this paper. It’s very necessary for dentin application.
• Figure 5: low resolution, please put another one.
• Novel references should be added.
• The description of SEM methodology needs improvement. Plus, the methodology needs major improvements.
• When defining the used abbreviations on the text, please use the full name first followed by (abbreviation), and not the other way around.
• Discussion should be more explicative and you shou add more about composition of adhesives and you should compare with novel studies. Discussion section needs further improvements. The hypothesis should be accepted or rejected after the presentation of results.
• Please add more limitations for this study
• Please use updated and recent papers in the literature review to give more sense to the reader.

---

## Round 0.2 · accepted · Accept

Dear Authors,

The revised manuscript is now suitable for publication. Once again, thank you very much for considering PeerJ to publish your work.

Reviewer 1 ·

Basic reporting

no comment

Experimental design

no comment

Validity of the findings

no comment

Additional comments

The authors improved the manuscript.